# Association of Serum Uric Acid with Diabetes in Premenopausal and Postmenopausal Women—A Prospective Cohort Study in Shanghai, China

**DOI:** 10.3390/ijerph192316137

**Published:** 2022-12-02

**Authors:** Qian Wu, Ying Guan, Chunze Xu, Na Wang, Xing Liu, Feng Jiang, Qi Zhao, Zhongxing Sun, Genming Zhao, Yonggen Jiang

**Affiliations:** 1Key Laboratory of Public Health Safety of Ministry of Education, School of Public Health, Fudan University, Shanghai 200032, China; 2Songjiang District Center for Disease Control and Prevention, Shanghai 201600, China; 3Department of Social Medicine, School of Public Health, Fudan University, Shanghai 200032, China

**Keywords:** cohort study, serum uric acid, diabetes, premenopausal women, postmenopausal women

## Abstract

There have been few prospective studies on the association between serum uric acid (SUA) and the risk of diabetes in women, and there have been few large-scale Chinese studies based on menopause to investigate the association. Therefore, the present study aimed to investigate the above relationship in Chinese female adults without diabetes. Methods: Data from 5743 premenopausal women and 11,287 postmenopausal women aged 20–74 years were obtained from the Shanghai Suburban Adult Cohort and Biobank (SSACB) study conducted in China. Cox regression models were applied to evaluate the association between SUA levels and the risk of diabetes. Restricted cubic spline analysis and stratified analysis on the basis of menopausal status were performed to explore the dose–response association between SUA levels and diabetes. Results: Among 17,030 participants, incidence rates of diabetes were 3.44/1000 person-years in premenopausal and 8.90/1000 person-years in postmenopausal women. The SUA levels in postmenopausal women were higher than that in premenopausal women (*p* < 0.0001). In Cox regression analysis, after adjusting for confounding factors, for each 10 µmol/L increase in SUA levels, the adjusted HR of diabetes was 1.01 (95% CI: 0.97–1.04) in postmenopausal women, and 1.03 (95% CI: 1.01–1.04) in premenopausal women. Compared with the lowest quartile of SUA levels, the HR (95% CI) of diabetes in the highest quartile was 0.99 (0.55–1.79) in premenopausal women and 1.39 (1.07–1.81) in postmenopausal women. Compared with those without hyperuricemia, the HR (95% CI) for diabetes was 1.89 (0.67–5.31) in premenopausal women with hyperuricemia, and 1.55 (1.19–2.02) in postmenopausal women. Moreover, restricted cubic splines models showed that there was a linear relationship between SUA levels and diabetes risk in premenopausal (*p* for nonlinear = 0.99) and postmenopausal women (*p* for nonlinear = 0.95). Furthermore, the restricted cubic spline graph showed that the risk of diabetes in postmenopausal women increased with an increase in SUA levels (*p* = 0.002). Conclusions: In a cohort of Chinese adult women, SUA levels are associated with diabetes risk in postmenopausal women, but this association was not observed in premenopausal women.

## 1. Introduction

Diabetes is a chronic disease that is characterized by disorders of blood glucose metabolism. According to the 2021 International Diabetes Federation [1], approximately 537 million adults aged 20–79 years had diabetes globally in 2021; thus, approximately 1 in 10 people had diabetes. By 2030, that number is expected to rise to 643 million, and by 2045, that number will rise to 783 million. The number of diabetics in China has reached 140 million, ranking first in the world, and the prevalence has risen from 10.9% in 2013 to 12.4% in 2018, according to a study [2].

Serum uric acid (SUA) is the end product of the metabolism of purine nucleotides. A growing number of epidemiological studies have shown that SUA is a risk factor for diabetes [3,4,5,6]. However, the relationship between SUA levels and diabetes risk is still controversial [7,8,9]. Studies have shown that the correlation between SUA levels and diabetes is sex-specific [6,7,8,9,10], whereas some studies have shown that there is a significant association in both men and women [10]; still, other studies suggest the association exists only in men [9], and others suggest that it only exists in women [6,7,8].

Although some studies have analyzed the relationship between SUA levels and the risk of diabetes in women [11,12,13], menopause is independently associated with SUA in women [14]. After menopause, female sex hormones decrease significantly, leading to an increase in SUA levels [15]. Whether the menopausal status of women affects the relationship between SUA levels and diabetes is still controversial. In a prospective study in China [16], the results of stratified analysis according to the general menopause time of Chinese women showed that there was no statistically significant association between SUA levels and prediabetes in women aged <48 years; meanwhile, there was a positive correlation between SUA levels and prediabetes in women aged ≥48 years. Unfortunately, the study did not analyze women’s true menopausal status. In the United States, results from the National Health and Nutrition Examination Survey [17] showed that SUA increased the risk of insulin resistance in premenopausal women, but had no effect on postmenopausal women. However, this was a cross-sectional study, and it had certain limitations. Few prospective studies have investigated the relationship between SUA levels and diabetes risk stratified by menopausal status, especially in Chinese women. Therefore, more prospective studies are needed to clarify the association between SUA levels and diabetes risk in women, and to determine whether this relationship is influenced by menopausal status. Therefore, we aimed to investigate the association between SUA levels and the future risk for incident diabetes in premenopausal and postmenopausal women in a population-based, well-established, prospective cohort.

## 2. Materials and Methods

### 2.1. Subjects

The research subjects were from the Shanghai Suburban Adult Cohort and Biobank (SSACB) study [18]. The cohort was established from April 2016 to December 2017, and the detailed methodology for this study has been described in previous studies [18]. Briefly, the cohort used a multi-stage stratified cluster sampling method to select four communities in Songjiang District, Shanghai, and collected health information from 31 neighborhood committees and 16 administrative villages in these four communities. In the end, we recruited a total of 21,621 women, all of whom were Shanghai natives or had lived in Shanghai for more than 5 years, aged between 20 and 74. Participants were excluded if they met one of the following criteria: participants with missing data for SUA, glucose profiles, or BMI at baseline; had diabetes before baseline investigation; had critical illnesses such as cancer, cirrhosis, chronic hepatitis, mental disorders, and chronic kidney disease. As a result, a total of 17,030 female participants were eligible subjects that were included in the study, after excluding 4591 participants (Figure 1). This project was approved by the Medical Research Ethics Committee of the School of Public Health of Fudan University (IRB number 2016-04-0586). All of the recruited participants provided written informed consent before the investigation.

### 2.2. Data Collection

The sociodemographic characteristics (age, sex, education level, marital status, etc.), lifestyle factors (current smoking status, current alcohol consumption, and physical activity), and chronic disease history (hypertension, diabetes, dyslipidemia, etc.) of the subjects were collected by trained interviewers through face-to-face interviews via a structured questionnaire. We assessed physical activity using the International Physical Activity Questionnaire. Current smoking status was defined as currently smoking more than 1 cigarette per day for more than 6 months, and was divided into yes or no. Current alcohol drinking status was defined as currently drinking more than three times per week for more than six months, and was classified as yes or no [19]. Height, weight, and blood pressure were measured using standard methods. Each subject was measured twice for height and weight, and the mean values were used to calculate BMI. Blood pressure was measured using an electronic sphygmomanometer three consecutive times, and the average value was taken for analysis.

### 2.3. Blood Collection Laboratory Measurements

All of the participants were asked to fast for at least 8 h before the start of the investigation, after which fasting venous blood was collected and measured at the DiAn Medical Laboratory Center. Total cholesterol (TC) levels were measured with enzymatic colorimetry (Roche Cobas C501 automatic biochemical analyzer); low-density lipoprotein cholesterol (LDL-C), high-density lipoprotein cholesterol (HDL-C), and triglyceride (TG) were measured using a colorimetric method (Roche Cobas C501 automatic biochemical analyzer). The fasting plasma glucose (FPG) levels were determined using the glucokinase method (Roche P800 biochemical analyzer). Serum creatinine (Scr) was determined with an enzymatic method (Roche Cobas C702 automatic biochemical analyzer), and SUA levels were measured using a colorimetric method (Roche Cobas C702 automatic biochemical analyzer). The hemoglobin A1c (HbA1c) levels were determined using high-pressure liquid chromatography (TOSOH G8 automatic biochemical analyzer).

### 2.4. Menopausal Status

We obtained the menopausal status and time of menopause of the study subjects through questionnaires, and then classified them as postmenopausal or premenopausal. Menopause was defined as the last menstrual period occurring ≥ 12 months before the interview [14].

### 2.5. Diabetes Assessment and Follow-up

We used the medical information system of the Shanghai Songjiang District Health Management Platform for follow-up, which imported the medical information of the study subjects every six months. Our follow-up outcome was diabetes events, which were collected according to the Diabetes Disease Registry and Reporting System (Electronic Medical Record (EMR)), using detailed records of disease diagnosis codes and time of disease diagnosis based on the subject’s unique identification number [20]. In this system, the coding of diabetes was based on the International Classification of Diseases, tenth edition (ICD-10), so the coding of diabetes in this study was determined as E10-E14 [21] at follow-up. Participants who self-reported diabetes at baseline, or who had a history of diabetes, were excluded to ensure that all participants had no history of diabetes before the cohort began. During subsequent follow-ups, the earliest onset of diabetes was recorded as the follow-up outcome. The duration of the cohort was from the baseline date to 31 January 2022.

### 2.6. Diagnostic Criteria

Hypertension was defined as SBP/DBP ≥ 140/90 mm Hg, or with a previous diagnosis history [22]. Hyperuricemia was defined as an SUA level > 360 µmol/L for women [23]. The estimated glomerular filtration rate (eGFR) was calculated using the Modification of Diet in Renal Disease (MDRD) Study Equation for the Chinese population [24]:

eGFR MDRD (mL/min/1.73 m^2^) = 175 × (serum creatinine (mg/dL))^−1.234^ × (age^−0.179^) × 0.79 (if female). The diagnosis of dyslipidemia was TC ≥ 6.22 mmol/L, or TG ≥ 2.26 mmol/L, or LDL-C ≥ 4.14 mmol/L, or HDL-C < 1.04 mmol/L, or previously diagnosed hyperlipidemia [25].

### 2.7. Quality Control

The implementation process and quality control of the Shanghai Suburban Adult Cohort and Biobank (SSACB) study were coordinated by the CDC of Songjiang District, and the School of Public Health of Fudan University. For each sub-district (town) where the onsite investigation was carried out, the community health service center was responsible for the implementation and quality control of the sub-district (town). A unified questionnaire and detailed rules for filling out the questionnaire were formulated. Before the investigation, the investigators were trained uniformly and strictly by the CDC doctor in the Songjiang District and by the on-site instructor of the School of Public Health of Fudan University, in order to ensure the quality of the investigation.

### 2.8. Statistical Analysis

Continuous variables were expressed as mean (SD) or median (interquartile range), and categorical variables were expressed as frequency (%). ANOVA, the Mann–Whitney U test, and the chi-square test were used to analyze continuous data, skewed continuous data, and categorical data, respectively.

Baseline SUA levels were divided into quartiles (quartiles 1–4): <214 µmol/L, 214–248 µmol/L, 249–289 µmol/L, and >289 µmol/L for premenopausal women, and <230 µmol/L, 230–266 µmol/L, 267–307 µmol/L, and >307 µmol/L for postmenopausal women.

Cox regression models were used to calculate hazard ratios (HRs) and 95% confidence intervals (CIs) between baseline SUA levels and diabetes. An increasing number of covariates were adjusted as follows: Model1 was adjusted for age, education, and marriage at baseline; Model2 was further adjusted for current smoking, current alcohol drinking, physical activity, and BMI at baseline; and Model3 was further adjusted for hypertension and dyslipidemia at baseline. Restricted cubic spline (RCS) analysis with three knots was applied to study the dose-response association between SUA levels and diabetes.

Analyses were performed using SPSS 26.0 software (IBM SPSS Inc., Chicago, IL, USA) and R version 4.1.2 (R Development Core Team, Vienna, Austria). A *p*-value < 0.05 was considered statistically significant, and all of the tests were two-sided.

## 3. Results

### 3.1. Demographic Characteristics of the Study Participants

At baseline, we included a total of 17,030 subjects, of whom 33.72% (*n* = 5743) were premenopausal women and 66.28% (*n* = 11,287) were postmenopausal women. The mean ages of the premenopausal and postmenopausal women were 42.19 ± 9.15 and 60.31 ± 6.67, respectively. During 83,600 person-years (median 5.00 years) of follow-up, 100 premenopausal women and 485 postmenopausal women developed diabetes; the incidence was 3.44/1000 person-years for premenopausal women and 8.90/1000 person-years for postmenopausal women. The baseline characteristics of premenopausal and postmenopausal women are shown in Table 1. At baseline, compared with premenopausal women, postmenopausal women were older, had fewer years of education, and had a higher prevalence of hypertension and dyslipidemia. Additionally, they had higher TG, TC, and LDL-C levels, but there was no significant difference in the HDL-C. Moreover, premenopausal women had significantly higher systolic and diastolic blood pressure; BMI, FPG, and HAb1C levels; and had lower eGFR (all *p* < 0.05). SUA levels in premenopausal women ranged from 290 to 576 µmol/L, with a mean level of 254.75 µmol/L; in postmenopausal women, it ranged from 308 to 653 µmol/L, with a mean level of 272.63 µmol/L. The levels were higher in postmenopausal women than in premenopausal women (Table 1).

### 3.2. Incidence Density of Diabetes in Premenopausal and Postmenopausal Women

Among premenopausal women, diabetes incidence density was higher in the third and fourth quantiles of SUA levels than in the first, but lower in the second quantile (*p* for trend = 0.021) (Figure 2A). Among postmenopausal women, the second, third, and fourth quantiles all had higher diabetes densities than the first (*p* for trend < 0.001) (Figure 2B).

### 3.3. Association between SUA Levels and Diabetes in Premenopausal and Postmenopausal Women

There were 100 (1.74%) premenopausal women who had diabetes events. After adjusting for confounders, the SUA levels (every 10 µmol/L increase) were not associated with the risk of diabetes events (HR: 1.01, 95% CI: 0.97–1.04). Compared with the first quartile of SUA levels, the risk of diabetes events in the second, third, and fourth quartiles of SUA levels did not increase, and the HRs (95% CI) were 0.85 (0.45–1.61), 0.96 (0.53–1.74), and 0.99 (0.55–1.79), respectively (Table 2). Moreover, there was no significant association between hyperuricemia and diabetes risk among premenopausal women (HR 1.89, 95% CI: 0.67–5.31) (Table 2).

There were 485 (4.30%) postmenopausal women who had diabetes events. After adjusting for confounders, for every 10 µmol/L increase in SUA levels, the risk of diabetes events increased by 3% (HR: 1.03, 95% CI: 1.01–1.04). Compared with the first quartile of SUA levels, the multivariate-adjusted HRs (95% CI) for diabetes were 1.06 (0.80–1.40) for the second, 1.10 (0.84–1.46) for the third, and 1.39 (1.07–1.81) for the fourth quartile of SUA levels. In addition, postmenopausal women with hyperuricemia had a 55% increased risk of developing diabetes compared with women without hyperuricemia (HR: 1.55, 95% CI: 1.19–2.02) (Table 2).

We explored the shape of the associations between SUA levels and the risk of diabetes, using the restricted cubic spline curve (Figure 3). There was a linear relationship between SUA levels and diabetes risk in premenopausal (*p* for nonlinear = 0.99) and postmenopausal women (*p* for nonlinear = 0.95). Furthermore, the restricted cubic spline graph shows that the risk of diabetes in postmenopausal women increased with an increase in SUA levels (*p* = 0.002), but the risk of diabetes did not change with SUA levels in premenopausal women (*p* = 0.846) (Figure 3).

## 4. Discussion

In this prospective study based on adult women in the Chinese community, we found a positive association between SUA levels and diabetes risk in postmenopausal women. This association was independent of age, education level, lifestyle factors (current smoking status, current alcohol consumption, and physical inactivity), BMI, and diabetes. However, baseline SUA levels in premenopausal women were not associated with diabetes events. In addition, there was a linear relationship between SUA levels and the risk of diabetes in premenopausal and postmenopausal women. The risk of diabetes increased with an increase in SUA levels in postmenopausal women, while the risk of diabetes in postmenopausal women did not change with a change in SUA levels. To the best of our knowledge, this is the first prospective study in China to stratify SUA levels and diabetes risk in women strictly on the basis of menopausal status. The results of this study are similar to those from a prospective study in Japan [26], but with a larger cohort and a linear relationship between SUA as a continuous variable and diabetes risk.

In recent years, studies have revealed that SUA levels are positively correlated with the risk of diabetes, especially in women [11,27]; however, the evidence for this link remains unclear. One possible explanation lies in estrogen and its clinical effects on women [16]. Estrogen in postmenopausal women gradually decreases, and possibly due to the accompanying decline in estrogen, SUA levels also generally become higher in postmenopausal women than they are in premenopausal women [28,29], which was also observed in our study. The explanation given by some studies for this phenomenon is that estrogen promotes the secretion of SUA during the reproductive period [30,31]. Therefore, it is necessary to stratify the risk of diabetes in premenopausal and postmenopausal women according to their menopausal status. However, there have been few studies in China that stratified women according to their menopausal status.

Similar results have been reported regarding premenopausal and postmenopausal differences in the relationship between SUA levels and diabetes risk. In a prospective study in China [16], the results of stratified analysis according to the general menopause time of Chinese women showed that there was no statistically significant association between SUA levels and prediabetes in women aged <48 years, while there was a positive correlation between SUA levels and prediabetes in women aged ≥48 years, but unfortunately the study did not analyze women’s true menopausal status, and stratified only by common menopause age, which is not accurate enough. A Korean cohort study [31] analyzed 1983 women according to their menopausal status, and the results showed that SUA in premenopausal women was not significantly associated with the risk of diabetes (HR = 8.01, P5% CI: 0.91–70.85, *p* = 0.061), but in postmenopausal women, the risk of diabetes was significantly increased in Q4 group compared with Q1 (HR = 7.26, 95% CI: 1.64–32.15, *p* = 0.009).

There are many studies that have explained the possible mechanisms between high SUA levels and the increased risk of diabetes [32,33,34]. Firstly, high SUA levels can lead to an impaired acetylcholine-induced vasodilation response, thereby impairing endothelial nitric oxide production and leading to endothelial cell dysfunction [35], a process that is associated with insulin resistance and the initiating process of diabetes [36,37]. In addition, high SUA levels can directly damage pancreatic β cells, leading to impaired pancreatic β cell function, reduced insulin synthesis, glucose metabolism disorder, and induced diabetes [38]. However, hyperinsulinemia due to insulin resistance increases SUA levels by reducing renal uric acid excretion, and accumulating uric acid-producing substrates [13,39,40]. Therefore, it is unclear whether hyperuricemia is a consequence of insulin resistance or a precursor to insulin resistance [27]. At present, numerous studies have shown that estrogen decline can cause an increase in SUA levels [41]. Reports have shown that endothelial function declines with estrogen deficiency [42]. These studies may explain a mechanism by which decreased estrogen levels lead to increased uric acid levels, which in turn may lead to endothelial dysfunction, and predisposes postmenopausal women to insulin resistance [16]. In addition, sex differences in glucose homeostasis have been observed in human and rodent models, suggesting that there is a potentially protective role of female sex hormones in preventing diabetes [43]. A large cohort study in China involving 300,000 people found that perimenopausal and postmenopausal women had an increased risk of developing diabetes compared to premenopausal women, with HRs of 1.17 (95% CI: 1.06–1.29) and 1.15 (1.06–1.25) [44]. Estradiol has been shown in animal experiments to increase insulin content and glucose-stimulated insulin secretion in mouse islets, suggesting that this hormone plays an active role in glucose homeostasis [45]. In women after menopause, this protective effect gradually disappears due to the decline in estrogen levels, which may lead to the occurrence of diabetes.

This study was a large prospective cohort study of community-based adults that involved a female population aged 20–74, with a large sample size and good representation. All biochemical indicators and the contents of the questionnaire were tested and investigated by professionals who were trained according to standard procedures, and the authenticity was good. We obtained women’s menopause status and menopause time through questionnaires instead of through a stratified analysis based on women’s common menopause age, which made the data more accurate. We also assessed the nonlinear relationship between SUA levels and diabetes risk in pre and postmenopausal women, which is rare in previous studies. With the increasing incidence of diabetes year by year, obtaining the cut-off value of SUA is more conducive to preventing and controlling diabetes.

There are some limitations in this study. Firstly, the outcome events were obtained through the medical information system, and there may have been some omissions, leading to the low incidence of diabetes. The low incidence of diabetes, especially among premenopausal women, somewhat reduced the statistical power of the analysis. Secondly, we did not measure estrogen levels in women, which may be related to SUA levels and diabetes in premenopausal women, and could help explore the underlying mechanisms and pathophysiological processes between SUA levels and diabetes. Thirdly, SUA is a dynamic indicator, yet we only measured SUA levels at baseline; frequent measurement of SUA levels can more accurately assess its impact on diabetes. Using only a single value of SUA may underestimate the association. In addition, we did not account for confounding factors, such as diet and genetics, which may have affected the validity of the conclusions. Moreover, our study was conducted in Songjiang District, Shanghai, China; whether the results can be generalized to other populations remains to be considered. Finally, in this study, the ICD coding range of diabetes used was E10–E14, so there was a lack of classification of diabetes subtypes. However, the age of the subjects in this study ranged from 20 to 74 years old, and in order to prevent omissions, we also used diabetes management data, which mainly records the incidence of type 2 diabetes in the community. Thus, we hypothesized that the majority of cases were type 2 diabetes.

## 5. Conclusions

In conclusion, our findings suggest that SUA levels are associated with diabetes risk in postmenopausal women, but this association was not observed in premenopausal women. Further studies are necessary in order to explore the pathogenic mechanism of this association. From the perspective of practice and prevention, this study suggests that SUA levels in postmenopausal women should regularly be checked, and the factors related to SUA elevation such as diet and lifestyle, should be adjusted in order to reduce the risk of diabetes. For postmenopausal women with high SUA levels, attention should be given to the prevention of diabetes.

## Figures and Tables

**Figure 1 ijerph-19-16137-f001:**
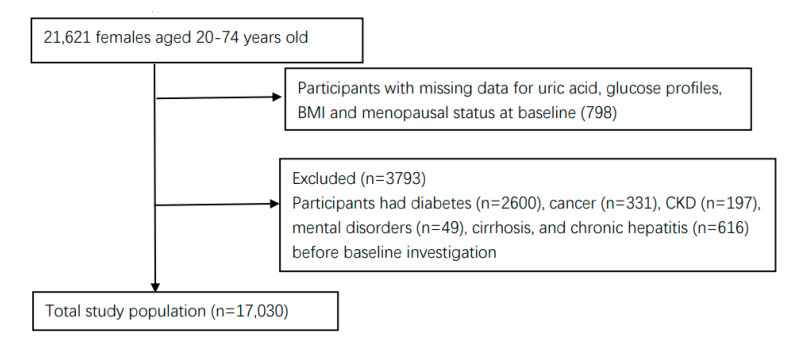
Selection process for the participants.

**Figure 2 ijerph-19-16137-f002:**
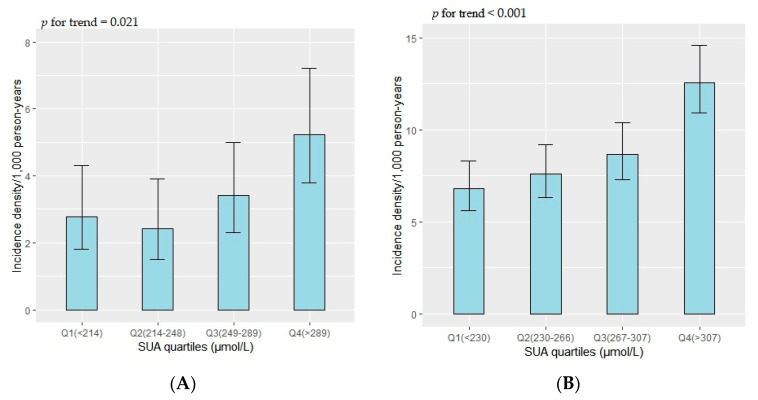
Incidence density of diabetes in premenopausal (**A**) and postmenopausal women (**B**).

**Figure 3 ijerph-19-16137-f003:**
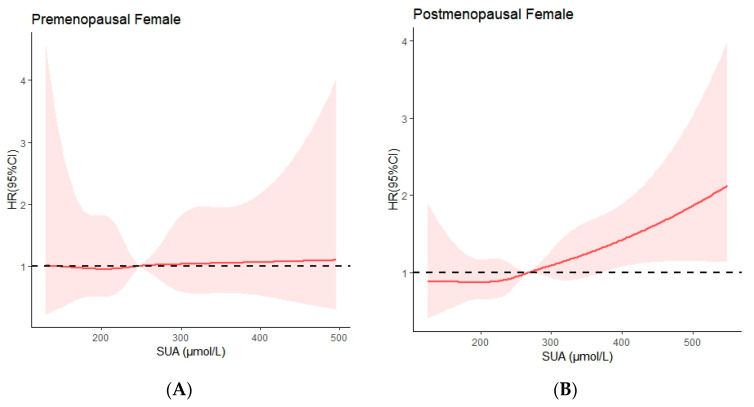
The relationship between SUA levels and risk of diabetes in premenopausal (**A**) and postmenopausal women (**B**). Adjusted cubic spline models show an association between SUA levels and the risk of diabetes in premenopausal (**A**) and postmenopausal women (**B**). Models were adjusted for age, education level, current smoking, current alcohol drinking, BMI, hypertension, and dyslipidemia. The solid line and pink shading represent the estimated hazards ratio and its 95% confidence interval. *p* nonlinear for premenopausal women = 0.99; *p* nonlinear for postmenopausal women = 0.95.

**Table 1 ijerph-19-16137-t001:** The baseline characteristics of premenopausal and postmenopausal women.

Baseline Characteristics	Premenopausal Women	Postmenopausal Women	*χ^2^/F/H*	*p*
n	5743	11,287		
Age (years, $\bar{x}$ ± s)	42.19 ± 9.15	60.31 ± 6.67	−147.12	<0.0001
BMI (kg/m^2^, $\bar{x}$ ± s)	23.07 ± 3.32	24.19 ± 3.20	−21.15	<0.0001
TG [mmol/L, $\bar{M}$ (Q1, Q3)]	1.08 (0.81,1.49)	1.38 (1.03,1.87)	631.32	<0.0001
LDL-C (mmol/L, $\bar{x}$ ± s)	2.55 ± 0.75	2.95 ± 0.84	−29.78	<0.0001
HDL-C (mmol/L, $\bar{x}$ ± s)	1.51 ± 0.34	1.50 ± 0.34	1.07	0.283
TC (mmol/L, $\bar{x}$ ± s)	4.64 ± 0.84	5.18 ± 0.92	−37.26	<0.0001
eGFR (mmol/L, $\bar{x}$ ± s)	118.92 ± 23.65	105.26 ± 26.82	32.66	<0.0001
FPG (mmol/L, M (Q1, Q3)]	4.57 (4.22,4.88)	4.68 (4.23,5.25)	105.19	<0.0001
HAb1C(mmol/L, $\bar{x}$ ± s)	5.37 ± 0.41	5.63 ± 0.40	−38.73	<0.0001
Blood pressure (mmHg, $\bar{x}$ ± s)				
Systolic	122.91 ± 17.97	135.84 ± 19.06	−42.25	<0.0001
Diastolic	75.87 ± 10.82	79.67 ± 9.98	−22.63	<0.0001
SUA range (µmol/L)	290–576	308–653		
SUA (µmol/L)	254.75 ± 57.94	272.63 ± 61.49	−18.29	<0.0001
Education (year, %)			4578.90	<0.0001
<7	930 (16.2)	7102 (62.9)		
7–9	2596 (45.2)	3266 (28.9)		
10–12	991 (17.3)	864 (7.7)		
>12	1226 (21.3)	55 (0.5)		
Marital status (%)			35.46	<0.0001
Married	5387 (93.8)	10,293 (91.2)		
Unmarried/Divorced/Widowed	356 (6.2)	994 (8.8)		
Current smoking (%)	40 (0.7)	149 (1.3)	13.49	<0.0001
Current alcohol drinking (%)	63(1.1)	162 (1.4)	3.34	0.068
Physical activity (%)	1899 (33.1)	3423 (30.3)	13.30	<0.0001
Hyperuricemia (%)	256 (4.5)	903 (8.0)	75.32	<0.0001
Hypertension (%)	1323 (23.0)	6170 (54.7)	1545.28	<0.0001
Dyslipidemia (%)	810 (14.1)	3104 (27.5)	385.92	<0.0001

**Table 2 ijerph-19-16137-t002:** Association between baseline SUA levels and risk of diabetes mellitus.

	Events/N (%)	Model1	Model2	Model3
Premenopausal women				
Q1	20/1414 (1.4)	1.00	1.00	1.00
Q2	18/1459 (1.2)	0.94 (0.50–1.78)	0.87 (0.46–1.64)	0.85 (0.45–1.61)
Q3	25/1457 (1.7)	1.26 (0.70–2.27)	1.01 (0.56–1.84)	0.96 (0.53–1.74)
Q4	37/1413 (2.6)	1.89 (1.09–3.27)	1.21 (0.68–2.16)	0.99 (0.55–1.79)
Hyperuricemia				
No	96/5680 (1.7)	1.00	1.00	1.00
Yes	4/63 (6.3)	4.17 (1.53–11.35)	2.69 (0.97–7.43)	1.89 (0.67–5.31)
SUA (every 10 µmol/L increase)		1.52 (1.16–1.99)	1.02 (0.99–1.06)	1.01 (0.97–1.04)
Postmenopausal women				
Q1	93/2801 (3.3)	1.00	1.00	1.00
Q2	105/2835 (3.7)	1.12 (0.85–1.49)	1.06 (0.80–1.41)	1.06 (0.80–1.40)
Q3	116/2784 (4.2)	1.27 (0.97–1.67)	1.14 (0.87–1.51)	1.10 (0.84–1.46)
Q4	171/2867 (6.0)	1.81 (1.40–2.32)	1.49 (1.15–1.94)	1.39 (1.07–1.81)
Hyperuricemia				
No	415/10384 (4.0)	1.00	1.00	1.00
Yes	70/903 (7.8)	1.92 (1.49–3.48)	1.64 (1.26–2.13)	1.55 (1.19–2.02)
SUA (every 10 µmol/L increase)		1.04 (1.03–1.06)	1.03 (1.02–1.04)	1.03 (1.01–1.04)

Model1: adjusted for age, education, and marriage; Model2: adjusted for Model1 covariates plus current smoking, current alcohol drinking, exercise, and BMI; Model3: adjusted for Model1 covariates plus hypertension and dyslipidemia. Baseline SUA levels were divided into quartiles (quartiles 1–4: <214 µmol/L, 214–248 µmol/L, 249–289 µmol/L, and >289 µmol/L for premenopausal women, and <230 µmol/L, 230–266 µmol/L, 267–307 µmol/L, and >307 µmol/L for postmenopausal women).

## Data Availability

All the data generated or analyzed during this study are available from the corresponding author upon reasonable request.

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
