# Peer review of "Association of Serum Uric Acid with Diabetes in Premenopausal and Postmenopausal Women—A Prospective Cohort Study in Shanghai, China"

_ijerph, 2022, doi:10.3390/ijerph192316137_

Round 1

Reviewer 1 Report

Thank you for the opportunity to review your manuscript titled “Association of uric acid with diabetes in premenopausal and postmenopausal women—A prospective cohort study in Shanghai, China”.

Abstract:

Some confusing results were reported, showing that there was no nonlinear relationship between serum uric acid and diabetes risk in either premenopausal (P for nonlinear = 0.99) or postmenopausal women (P for nonlinear = 0.95). why did the author report that serum uric acid is associated with diabetes risk in postmenopausal women? Please clarify it.

Introduction:

Please unify the abbreviation of “SUA”, some words are “Serum uric acid”.

Methods:

 The abbreviations of “LDL-C, HDL-C…” should write the full name in the first appearance.

Results:

“The restricted cubic spline graph shows that the risk of diabetes in postmenopausal women increases with the increase of serum uric acid (Figure 3.). ” Did the result have significance? Please clarify it.

What’s meaning of Q1-Q4 in Table 2? You should explain it in the methods and below the table.

Discussion:

What’s the significant findings in your study? authors should summarize it clearly and explain that what’s the findings different from previous study.  

Author Response

RE:  ijerph-2038791

Title: Association of serum uric acid with diabetes in premenopausal and postmenopausal women—A prospective cohort study in Shanghai, China 

Dear Reviewer:

Thank you very much for your comments. We really appreciate the your comments, which are very informative, suggestive and helpful. Based on the your comments, we have revised the manuscript. Changes to the manuscript were made using green font to facilitate the review process. Attached is the response to your comments, please check the attachment. We have used an editing service and have had native English speakers make edits and changes to our English language. We hope that the revision is acceptable.

Sincerely yours,

Qian Wu

Reviewer 2 Report

1. There needs to be an expansion of the limitations section to comment on threats to validity and reliability.

2. Another major section needed on implications for practice, research, and prevention.

Author Response

RE:  ijerph-2038791

Title: Association of serum uric acid with diabetes in premenopausal and postmenopausal women—A prospective cohort study in Shanghai, China 

Dear Reviewer:

Thank you very much for your comments. We really appreciate your comments, which are very informative, suggestive and helpful. Based on your comments, we have revised the manuscript. Changes to the manuscript were made using green font to facilitate the review process. Attached is the response to your comments, please check the attachment. We have used an editing service and have had native English speakers make edits and changes to our English language. We hope that the revision is acceptable.

Sincerely yours,

Qian Wu 

Reviewer 3 Report

General comment: this seems to be an exact  re-run to the of this paper missing from the reference list  in a much larger cohort,  and for Chinese as opposed to Japanese women.

https://doi.org/10.1097/gme.0000000000002035

Validation of prior smaller studies with much bigger cohorts is fine  - if this is what this paper is - but this emphasis should appear somewhere clearly. In the title? 

Is there  another message in the paper about approaches to analysis of data? Linear regression (this can be multi-factorial as well ( versus cubic spline type of regression analysis (single and multi-factorial). 

Title

What type of diabetes? 

Abstract

line 14: what type of diabetes? 

Line 25: rephrase

Lines 29 - 33: shorten 

Introduction

General comment: there is a lot of published work on the hormonal basis to lowered plasma uric acid and progesterone/estrogen levels in post-menopausal women e.g. 

https://ard.bmj.com/content/76/Suppl_2/1037.3

This needs covering in  the Introduction. 

Line 42: which types of diabetes are you covering?

Results 

Results reasonably well analysed and presented. 

Fig 3. I assume that the pink shaded area in Fig 3 in an illustration of variation in a composite index of the factors influencing the  hazard ratio index,   and how this composite index varies with SUA? 

The  solid and dashed lines indicate the trend between Hazard Ratio and serum uric acid concentration, weighted for these factors, as described in the title.   The pink shading needs explaining anyway. 

Hazard ratio is not diabetes as such but an composite index of risk factors associated with diabetic status. 

Discussion

General comment: Discussion reasonably well written an relevant. 

Is hormone replacement therapy and its effects on reducing uric acid accumulation in post menopausal women worth mentioning ?

https://ard.bmj.com/content/76/Suppl_2/1037.3 

Author Response

RE:  ijerph-2038791

Title: Association of serum uric acid with diabetes in premenopausal and postmenopausal women—A prospective cohort study in Shanghai, China 

Dear Reviewer:

Thank you very much for your comments. We really appreciate your comments, which are very informative, suggestive, and helpful. Based on your comments, we have revised the manuscript. Changes to the manuscript were made using a green font to facilitate the review process. Attached is the response to your comments, please check the attachment. We have used an editing service and have had native English speakers make edits and changes to our English language. We hope that the revision is acceptable.

Sincerely yours,

Qian Wu 

Reviewer 4 Report

Diabetes is a chronic disease associated with dysregulated glucose metabolism in the blood. Previously it was reported that the serum uric acid (SUA) was associated with increased risk of diabetes. Recent study has reported that the women are more prone to developing diabetes during menopause transition and post menopause stage. In the present study authors highlight the association of serum uric acid with diabetes in premenopausal and postmenopausal women in shanghai China. This study found that the risk of diabetes was positively associated with SUA levels in postmenopausal women. But in premenopausal women the baseline SUA was not associated with the risk of diabetes. I consider this manuscript entitled association of uric acid with diabetes in premenopausal and postmenopausal women –A prospective cohort study in Shanghai China for the publication in the journal of Environmental research and public health with minor revision.

1.     In-between pre and post menopause there is peri-menopause period of 12 month in which hormonal deregulation is observed. It would be good to know the level of SUA and its association with risk of diabetes. I hope authors have the data and they may consider including these women’s and consider analyzing SUA in Pre, Peri and Post menopause women’s and its association with diabetes risk.

2.     Number of subject (n) and % in Q1, Q2, Q3, Q4 should be included in table 

3.     Serum Uric acid was mentioned as SUA at some place and UA at some (including title), it is recommended to use single abbreviation throughout the manuscript so that readers of this article will not get confused.

4.     Below reference which associates the risk of diabetes with respect to menopause status should be cited

Wang, M., Gan, W., Kartsonaki, C. et al. Menopausal status, age at natural menopause and risk of diabetes in China: a 10-year prospective study of 300,000 women. Nutr Metab (Lond) 19, 7 (2022). https://doi.org/10.1186/s12986-022-00643-x

5. In abstract line 14, first sentence  not able to understand what author wants to say (Diabetes in women without diabetes).

6. SUA level is dynamic, and it would be highly recommended to have repeated analysis SUA level to determine its association with diabetes risk.

Author Response

(The authors gave the same response as above.)

Round 2

Reviewer 1 Report

Dear authors,

Please revised your manuscript according to the minor suggestions below:

Line 227: We explored the shape of the associations between SUA and the risk of diabetes...  please add the levels behind the "SUA", and ensure that all "SUA levels" are expressed consistently throughout the manuscript. 

Line 253-254: In addition, there was a linear relationship between SUA level and the risk of diabetes in premenopausal and postmenopausal women. Please change the "level" to "levels". 

Thanks!

Reviewer 3 Report

A much improved report 

Now ready for publication